# Houdini: Fooling Deep Structured Visual and Speech Recognition Models with Adversarial Examples

**Moustapha Cisse**
Facebook AI Research
moustaphacisse@fb.com

**Yossi Adi***
Bar-Ilan University, Israel
yossiadidrum@gmail.com

**Natalia Neverova***
Facebook AI Research
nneverova@fb.com

**Joseph Keshet**
Bar-Ilan University, Israel
jkeshet@cs.biu.ac.il

## Abstract

Generating adversarial examples is a critical step for evaluating and improving the robustness of learning machines. So far, most existing methods only work for classification and are not designed to alter the true performance measure of the problem at hand. We introduce a novel flexible approach named Houdini for generating adversarial examples specifically tailored for the final performance measure of the task considered, be it *combinatorial and non-decomposable*. We successfully apply Houdini to a range of applications such as speech recognition, pose estimation and semantic segmentation. In all cases, the attacks based on Houdini achieve higher success rate than those based on the traditional surrogates used to train the models while using a less perceptible adversarial perturbation.

## 1 Introduction

Deep learning has redefined the landscape of machine intelligence [22] by enabling several breakthroughs in notoriously difficult problems such as image classification [20, 16], speech recognition [2], human pose estimation [35] and machine translation [4]. As the most successful models are permeating nearly all the segments of the technology industry from self-driving cars to automated dialog agents, it becomes critical to revisit the evaluation protocol of deep learning models and design new ways to assess their reliability beyond the traditional metrics. Evaluating the robustness of neural networks to adversarial examples is one step in that direction [32]. Adversarial examples are synthetic patterns carefully crafted by adding a peculiar noise to legitimate examples. They are indistinguishable from the legitimate examples by a human, yet they have demonstrated a strong ability to cause catastrophic failure of state of the art classification systems [12, 25, 21]. The existence of adversarial examples highlights a potential threat for machine learning systems at large [28] that can limit their adoption in security sensitive applications. It has triggered an active line of research concerned with understanding the phenomenon [10, 11], and making neural networks more robust [29, 7] .

Adversarial examples are crucial for reliably evaluating and improving the robustness of the models [12]. Ideally, they must be generated to alter the task loss unique to the application considered directly. For instance, an adversarial example crafted to attack a speech recognition system should be designed to *maximize* the word error rate of the targetted system. The existing methods for generating adversarial examples exploit the gradient of a given differentiable loss function to guide the search in the neighborhood of legitimates examples [12, 25]. Unfortunately, the task loss of several structured prediction problems of interest is a combinatorial non-decomposable quantity that is not amenable

---

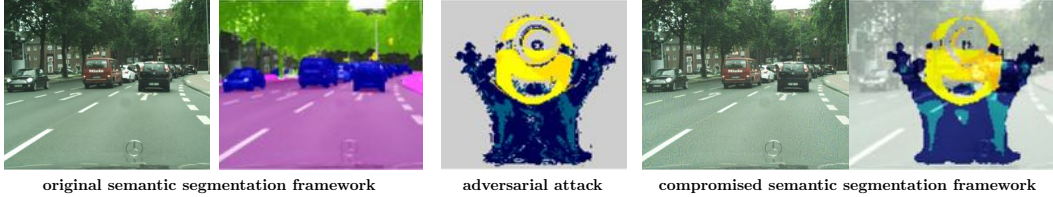

<div align="center">
original semantic segmentation framework     adversarial attack     compromised semantic segmentation framework
</div>

Figure 1: We cause the network to generate a *minion* as segmentation for the adversarially perturbed version of the original image. Note that the original and the perturbed image are indistinguishable.

to gradient-based methods for generating adversarial example. For example, the metric for evaluating human pose estimation is the (normalized) *percentage of correct keypoints*. Automatic speech recognition systems are assessed using their *word (or phoneme) error rate*. Similarly, the quality of a semantic segmentation is measured by the *intersection over union* (IOU) between the ground truth and the prediction. All these evaluation measures are non-differentiable.

The solutions for this obstacle in supervised learning are of two kinds. The first route is to use a consistent differentiable surrogate loss function in place of the task loss [5]. That is a surrogate which is guaranteed to converge to the task loss asymptotically. The second option is to directly optimize the task loss by using approaches such as Direct Loss Minimization [14]. Both of these strategies have severe limitations. (1) The use of differentiable surrogates is satisfactory for classification because the relationship between such surrogates and the classification accuracy is well established [34]. The picture is different for the above-mentioned structured prediction tasks. Indeed, there is no known consistency guarantee for the surrogates traditionally used in these problems (e.g. the connectionist temporal classification loss for speech recognition) and designing a new surrogate is nontrivial and problem dependent. At best, one can only expect a high positive correlation between the proxy and the task loss. (2) The direct minimization approaches are more computationally involved because they require solving a computationally expensive loss augmented inference for each parameter update. Also, they are notoriously sensitive to the choice of the hyperparameters. Consequently, it is harder to generate adversarial examples for structured prediction problems as it requires significant domain expertise with little guarantee of success when surrogate does not tightly approximate the task loss.

**Results.** In this work we introduce Houdini, the first approach for fooling any gradient-based learning machine by generating adversarial examples directly tailored for the task loss of interest be it combinatorial or non-differentiable. We show the tight relationship between Houdini and the task loss of the problem considered. We present the first successful attack on a deep Automatic Speech Recognition (ASR) system, namely a DeepSpeech-2 based architecture [1], by generating adversarial audio files not distinguishable from legitimate ones by a human (as validated by an ABX experiment). We also demonstrate the transferability of adversarial examples in speech recognition by fooling Google Voice™ in a black box attack scenario: an adversarial example generated with our model and not distinguishable from the legitimate one by a human leads to an invalid transcription by the Google Voice application (see Figure 8). We also present the first successful untargeted and targetted attacks on a deep model for human pose estimation [26]. Similarly, we validate the feasibility of untargeted and targetted attacks on a semantic segmentation system [38] and show that we can make the system hallucinate an arbitrary segmentation of our choice for a given image. Figure 1 shows an experiment where we cause the network to hallucinate a *minion*. In all cases, our approach generates better quality adversarial examples than each of the different surrogates (expressly designed for the model considered) without additional computational overhead thanks to the analytical gradient of Houdini.

## 2 Related Work

**Adversarial examples.** The empirical study of Szegedy et al. [32] first demonstrated that deep neural networks could achieve high accuracy on previously unseen examples while being vulnerable to small adversarial perturbations. This finding has recently aroused keen interest in the community [12, 28, 32, 33]. Several studies have subsequently analyzed the phenomenon [10, 31, 11] and various approaches have been proposed to improve the robustness of neural networks [29, 7]. More closely

related to our work are the different proposals aiming at generating better adversarial examples [12, 25]. Given an input (train or test) example $(x, y)$, an adversarial example is a perturbed version of the original pattern $\tilde{x} = x + \delta_x$ where $\delta_x$ is small enough for $\tilde{x}$ to be undistinguishable from $x$ by a human, but causes the network to predict an incorrect target. Given the network $g_\theta$ (where $\theta$ is the set of parameters) and a $p$-norm, the adversarial example is formally defined as:

$$\tilde{x} = \operatorname*{argmax}_{\tilde{x}: \|\tilde{x}-x\|_p \leq \epsilon} \ell\big(g_\theta(\tilde{x}), y\big) \tag{1}$$

where $\epsilon$ represents the strength of the adversary. Assuming the loss function $\ell(\cdot)$ is differentiable, Shaham et al. [31] propose to take the first order taylor expansion of $x \mapsto \ell(g_\theta(x), y)$ to compute $\delta_x$ by solving the following simpler problem:

$$\tilde{x} = \operatorname*{argmax}_{\tilde{x}: \|\tilde{x}-x\|_p \leq \epsilon} \big(\nabla_x \ell(g_\theta(x), y)\big)^T (\tilde{x} - x) \tag{2}$$

When $p = \infty$, then $\tilde{x} = x + \epsilon \text{sign}(\nabla_x \ell(g_\theta(x), y))$ which corresponds to the *fast gradient sign method* [12]. If instead $p = 2$, we obtain $\tilde{x} = x + \epsilon \nabla_x \ell(g_\theta(x), y)$ where $\nabla_x \ell(g_\theta(x), y)$ is often normalized. Optionally, one can perform more iterations of these steps using a smaller norm. This more involved strategy has several variants [25]. These methods are concerned with generating adversarial examples assuming a differentiable loss function $\ell(\cdot)$. Therefore they are not directly applicable to the task losses of interest. However, they can be used in combination with our proposal which derives a consistent approximation of the task loss having an analytical gradient.

**Task Loss Minimization.** Recently, several works have focused on directly minimizing the task loss. In particular, McAllester et al. [24] presented a theorem stating that a certain perceptron-like learning rule, involving feature vectors derived from loss-augmented inference, directly corresponds to the gradient of the task loss. While this algorithm performs well in practice, it is extremely sensitive to the choice of its hyper-parameter and needs two inference operations per training iteration. Do et al. [9] generalized the notion of the ramp loss from binary classification to structured prediction and proposed a tighter bound to the task loss than the structured hinge loss. The update rule of the structured ramp loss is similar to the update rule of the direct loss minimization algorithm, and similarly it needs two inference operations per training iteration. Keshet et al. [19] generalized the notion of the binary probit loss to the structured prediction case. The probit loss is a surrogate loss function naturally resulted in the PAC-Bayesian generalization theorems. it is defined as follows:

$$\bar{\ell}_{probit}(g_\theta(x), y) = \mathbb{E}_{\epsilon \sim \mathcal{N}(\mathbf{0}, \boldsymbol{I})} \left[ \ell(y, g_{\theta + \epsilon}(x)) \right] \tag{3}$$

where $\epsilon \in \mathbb{R}^d$ is a $d$-dimensional isotropic Normal random vector. [18] stated finite sample generalization bounds for the structured probit loss and showed that it is strongly consistent. Strong consistency is a critical property of a surrogate since it guarantees the tight relationship to the task loss. For instance, an attacker of a given system can expect to deteriorate the task loss if she deteriorates the consistent surrogate of it. The gradient of the structured probit loss can be approximated by averaging over samples from the unit-variance isotropic normal distribution, where for each sample an inference with perturbed parameters is computed. Hundreds to thousands of inference operations are required per iteration to gain stability in the gradient computation. Hence the update rule is computationally prohibitive and limits the applicability of the structured probit loss despite its desirable properties.

We propose a new loss named Houdini. It shares the desirable properties of the structured probit loss while not suffering from its limitations. Like the structured probit loss and unlike most surrogates used in structured prediction (e.g. structured hinge loss for SVMs), it is tightly related to the task loss. Therefore it allows to reliably generate adversarial examples for a given task loss of interest. Unlike the structured probit loss and like the smooth surrogates, it has an analytical gradient hence require only a single inference in its update rule. The next section presents the details of our proposal.

## 3 Houdini

Let us consider a neural network $g_\theta$ parameterized by $\theta$ and the task loss of a given problem $\ell(\cdot)$. We assume $\ell(y, y) = 0$ for any target $y$. The score output by the network for an example $(x, y)$ is $g_\theta(x, y)$ and the network's decoder predicts the highest scoring target:

$$\hat{y} = y_\theta(x) = \operatorname*{argmax}_{y \in \mathcal{Y}} g_\theta(x, y). \tag{4}$$

Using the terminology of section 2, finding an adversarial example fooling the model $g_\theta$ with respect to the task loss $\ell(\cdot)$ for a chosen p-norm and noise parameter $\epsilon$ boils down to solving:

$$\tilde{x} = \operatorname*{argmax}_{\tilde{x}:\|\tilde{x}-x\|_p \leq \epsilon} \ell\big(y_\theta(\tilde{x}), y\big) \tag{5}$$

The task loss is often a combinatorial quantity which is hard to optimize, hence it is replaced with a differentiable *surrogate loss*, denoted $\bar{\ell}(y_\theta(\tilde{x}), y)$. Different algorithms use different surrogate loss functions: structural SVM uses the structured hinge loss, Conditional random fields use the log loss, etc. We propose a surrogate named Houdini and defined as follows for a given example $(x, y)$:

$$\bar{\ell}_H(\theta, x, y) = \mathbb{P}_{\gamma \sim \mathcal{N}(0,1)} \Big[ g_\theta(x, y) - g_\theta(x, \hat{y}) < \gamma \Big] \cdot \ell(\hat{y}, y) \tag{6}$$

In words, Houdini is a product of two terms. The first term is a stochastic margin, that is the probability that the difference between the score of the actual target $g_\theta(x, y)$ and that of the predicted target $g_\theta(x, \hat{y})$ is smaller than $\gamma \sim \mathcal{N}(0, 1)$. It reflects the confidence of the model in its predictions. The second term is the task loss, which given two targets is independent of the model and corresponds to what we are ultimately interested in maximizing. Houdini is a lower bound of the task loss. Indeed denoting $\delta g(y, \hat{y}) = g_\theta(x, y) - g_\theta(x, \hat{y})$ as the difference between the scores assigned by the network to the ground truth and the prediction, we have $\mathbb{P}_{\gamma \sim \mathcal{N}(0,1)}(\delta g(y, \hat{y}) < \gamma)$ is smaller than 1. Hence when this probability goes to 1, or equivalently when the score assigned by the network to the target $\hat{y}$ grows without a bound, Houdini converges to the task loss. This is a unique property not enjoyed by most surrogates used in the applications of interest in our work. It ensures that Houdini is a good proxy of the task loss for generating adversarial examples.

We can now use Houdini in place of the task loss $\ell(\cdot)$ in the problem 5. Following 2, we resort to a first order approximation which requires the gradient of Houdini with respect to the input $x$. The latter is obtained following the chain rule:

$$\nabla_x \big[ \bar{\ell}_H(\theta, x, y) \big] = \frac{\partial \bar{\ell}_H(\theta, x, y)}{\partial g_\theta(x, y)} \frac{\partial g_\theta(x, y)}{\partial x} \tag{7}$$

To compute the RHS of the above quantity, we only need to compute the derivative of Houdini with respect to its input (the output of the network). The rest is obtained by backpropagation. The derivative of the loss with respect to the network's output is:

$$\nabla_g \Big[ \mathbb{P}_{\gamma \sim \mathcal{N}(0,1)} \big[ g_\theta(x, y) - g_\theta(x, \hat{y}) < \gamma \big] \ell(y, \hat{y}) \Big] = \nabla_g \left[ \frac{1}{\sqrt{2\pi}} \int_{\delta g(y, \hat{y})}^{\infty} e^{-v^2/2} dv\, \ell(y, \hat{y}) \right] \tag{8}$$

Therefore, expanding the right hand side and denoting $C = 1/\sqrt{2\pi}$ we have:

$$\nabla_g \big[ \bar{\ell}_H(\hat{y}, y) \big] = \begin{cases} -C \cdot e^{-|\delta g(y, \hat{y})|^2/2} \ell(y, \hat{y}), & g = g_\theta(x, y) \\ C \cdot e^{-|\delta g(y, \hat{y})|^2/2} \ell(y, \hat{y}), & g = g_\theta(x, \hat{y}) \\ 0, & \text{otherwise} \end{cases} \tag{9}$$

Equation 9 provides a simple analytical formula for computing the gradient of Houdini with respect to its input, hence an efficient way to obtain the gradient with respect to the input of the network $x$ by backpropagation. The gradient can be used in combination with any gradient-based adversarial example generation procedure [12, 25] in two ways, depending on the form of attack considered. For an untargeted attack, we want to change the prediction of the network without preference on the final prediction. In that case, any alternative target $y$ can be used (e.g. the second highest scorer as the target). For a targetted attack, we set the $y$ to be the desired final prediction. Also note that, when the score of the predicted target is very close to that of the ground truth (or desired target), that is when $\delta g(y, \hat{y})$ is small as we expect from the trained network we want to fool, we have $e^{-|\delta g(y, \hat{y})|^2/2} \simeq 1$. In the next sections, we show the effectiveness of the proposed attack scheme on human pose estimation, semantic segmentation and automatic speech recognition systems.

## 4 Human Pose Estimation

We evaluate the effectiveness of Houdini loss in the context of adversarial attacks on neural models for human pose estimation. Compromising performance of such systems can be desirable for

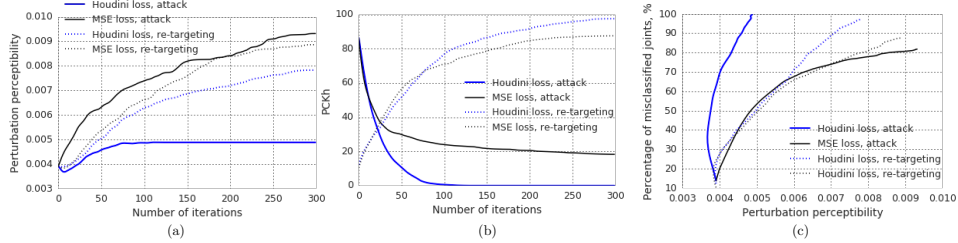

Figure 2: Convergence dynamics for pose estimation attacks: (a) perturbation perceptibility vs nb. iterations, (b) PCKh$^{0.5}$ vs nb. iterations, (c) proportion of re-positioned joints vs perceptibility.

manipulating surveillance cameras, altering the analysis of crime scenes, disrupting human-robot interaction or fooling biometrical authentication systems based on gate recognition. The pose estimation task is formulated as follows: given a single RGB image of a person, determine correct 2D positions of several pre-defined keypoints which typically correspond to skeletal joints. In practice, the performance is measured by the percentage of correctly detected keypoints (PCKh) (i.e. whose predicted locations are within a certain distance from the corresponding target positions) [3]:

$$\text{PCKh}^\alpha = \frac{\sum_{i=1}^N \mathbb{1}(\|y_i - \hat{y}_i\| < \alpha h)}{N}, \tag{10}$$

where $\hat{y}$ and $y$ are the predicted and the desired positions of a given joint respectively, $h$ is the head size of the person (known at test time), $\alpha$ is a threshold (set to $0.5$), and $N$ is the number of annotated keypoints. Pose estimation is a good example of a problem where we observe a discrepancy between the training objective and the final evaluation measure. Instead of directly minimizing the percentage of correctly detected keypoints, state-of-the-art methods rely upon a dense prediction of heatmaps, i.e. estimation of probabilities of every pixel corresponding to each of keypoint locations. These models can be trained with binary cross entropy [6], softmax [15] or MSE losses [26] applied to every pixel in the output space, separately for each plane corresponding to every keypoint. In our first experiment, we attack a state-of-the-art model for single person pose estimation based on Hourglass networks [26] and aim to minimize the value of PCKh$^{0.5}$ metric given the minimal perturbation. For this task we choose $\hat{y}$ as:

$$\hat{y} = \underset{\tilde{y}:\|\tilde{p}-p\|>\alpha h}{\text{argmax}} \; g_\theta(x, \tilde{y}) \tag{11}$$

where $p$ is the pixel coordinate on the heatmap corresponding to the argmax value of vector $y$. We perform the optimization iteratively till convergence with an update rule $\epsilon \cdot \frac{\nabla_x}{\|\nabla_x\|}$ where $\nabla_x$ are gradients with respect to the input and $\epsilon = 0.1$. We perform the evaluations on the validation subset of MPII dataset [3] consisting of 3000 images and defined as in [26]. We evaluate the perceived degree of perturbation where perceptibility is expressed as $\left(\frac{1}{n}\sum(x_i' - x_i)^2\right)^{1/2}$, where $x$ and $x'$ are original and distorted images, and $n$ is the number of pixels [32]. In addition, we report the structural similarity index (SSIM) [36] which is known to correlate well with the visual perception of image *structure* by a human. Figure 2 shows that Houdini only requires 100 iterations to maximally deteriorate the percentage of correct key-points from $89.4$ down to $0.57$ while MSE deteriorates the performance to only $24.12$ after 100 iterations. This observation underlines the importance of the loss function used to generate adversarial examples in structured prediction problems. Also, for untargeted attacks optimized to convergence, the perturbation generated with Houdini is up to $50\%$ less perceptible than the one obtained with MSE.

In the second experiment, we perform a targeted attack in the form of pose transfer, i.e. we force the network to hallucinate an arbitrary pose (with success defined, as before, given the target metric PCKh$^{0.5}$). The experimental setup is as follows: for a given pair of images $(i, j)$ we force the network to output the ground truth pose of the picture $i$ when the input is image $j$ and vice versa. This task is more challenging and depends on the similarity between the original and target poses. Surprisingly, targetted attacks are still feasible even when the two ground truth poses are very different. Figure 3 shows an example where the model predicts the pose of a human body in horizontal position for an adversarially perturbed image depicting a standing person (and vice versa). A similar experiment with two persons in standing and sitting positions respectively is also shown in Figure 3.

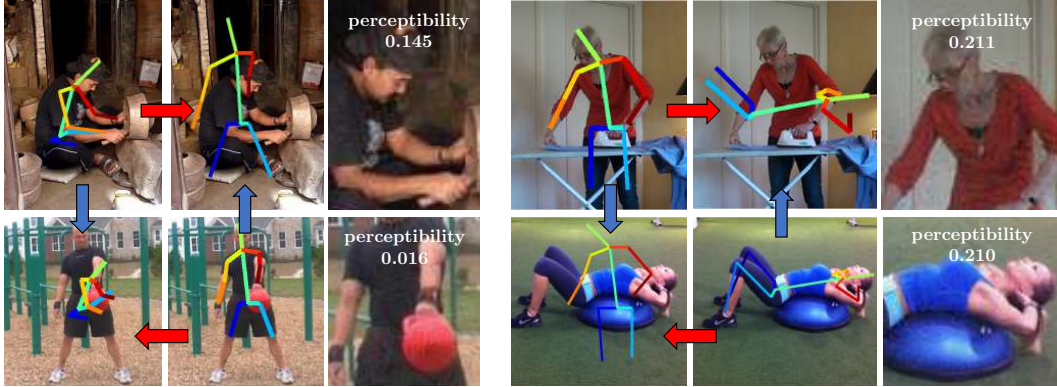

Figure 3: Examples of successful targetted attacks on a pose estimation system. Despite the important difference between the images selected, it is possible to make the network predict the wrong pose by adding an imperceptible perturbation. The images are part of the MPI dataset.

| Method | SSIM | | Perceptibility | |
|---|---|---|---|---|
| | @mIoU/2 | @mIoU$^{\text{lim}}$ | @mIoU/2 | @mIoU$^{\text{lim}}$ |
| untargeted: NLL loss | 0.9989 | 0.9950 | 0.0037 | 0.0117 |
| untargeted: Houdini loss | 0.9995 | 0.9959 | 0.0026 | 0.0095 |
| targetted: NLL loss | 0.9972 | 0.9935 | 0.0074 | 0.0389 |
| targetted: Houdini loss | 0.9975 | 0.9937 | 0.0054 | 0.0392 |

Table 1: Comparison of targetted and untargeted adversarial attacks on segmentation systems. mIoU/2 denotes 50% performance drop according to the target metric and mIoU$^{\text{lim}}$ corresponds to convergence or termination after 300 iterations. SSIM: the higher, the better, perceptibility: the lower, the better. Houdini based attacks are less perceptible.

## 5 Semantic segmentation

Semantic segmentation uses another customized metric to evaluate performance, namely the mean Intersection over Union (mIoU) measure defined as averaging over classes the IoU $=$ $\text{TP}/(\text{TP} + \text{FP} + \text{FN})$, where TP, FP and FN stand for true positive, false positive and false negative respectively, taken separately for each class. Compared to per-pixel accuracy, which appears to be overoptimistic on highly unbalanced datasets, and per-class accuracy, which under-penalizes false alarms for non-background classes, this metric favors accurate object localization with tighter masks (in instance segmentation) or bounding boxes (in detection). The models are trained with a per-pixel softmax or multi-class cross entropy losses depending on the task formulation, i.e. optimized for mean per-pixel or per-class accuracy instead of mIoU. Primary targets of adversarial attacks in this group of applications are self-driving cars and robots. Xie et al. [37] have previously explored adversarial attacks in the context of semantic segmentation. However, they exploited the same proxy used for training the network. We perform a series of experiments similar to the ones described in Sec. 4. That is, we show targetted and untargeted attacks on a semantic segmentation model. We use a pre-trained Dilation10 model for semantic segmentation [38] and evaluate the success of the attacks on the validation subset of Cityscapes dataset [8]. In the first experiment, we directly alter the target mIoU metric for a given test image in both targetted and untargeted attacks. As shown in Table 1, Houdini allows fooling the model at least as well as the training proxy (NLL) while using a less perceptible. Indeed, Houdini based adversarial perturbations generated to alter the performance of the model by $50\%$ are about $30\%$ less noticeable than the noise created with NLL.

The second set of experiments consists of targetted attacks. That is, altering the input image to obtain an arbitrary target segmentation map as the network response. In Figure 4, we show an instance of such attack in a segmentation transfer setting, i.e. the target segmentation is the ground truth segmentation of a different image. It is clear that this type of attack is still feasible with a small

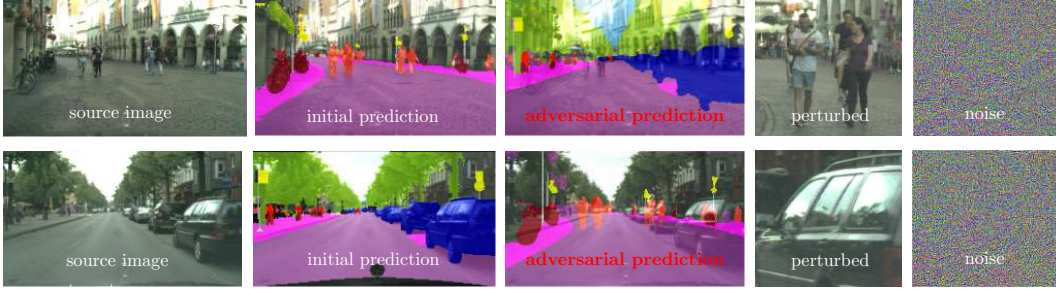

Figure 4: Targetted attack on a semantic segmentation system: switching target segmentation between two images from Cityscapes dataset [8]. The last two columns are respectively zoomed-in parts of the perturbed image and the adversarial perturbation added to the original one.

adversarial perturbation (even when after zooming in the picture). Figure 1 depicts a more challenging scenario where the target segmentation is an arbitrary map (e.g. a *minion*). Again, we can make the network hallucinate the segmentation of our choice by adding a barely noticeable perturbation.

# 6   Speech Recognition

We evaluate the effectiveness of Houdini concerning adversarial attacks on an Automatic Speech Recognition (ASR) system. Traditionally, ASR systems are composed of different components, (e.g. acoustic model, language model, pronunciation model, etc.) where each component is trained separately. Recently, ASR research has focused on deep learning based end-to-end models. These type of models get as input a speech segment and output a transcript with no additional post-processing. In this work, we use a deep neural network as our model with similar architecture to the one presented by [2]. The system is composed of two convolutional layers, followed by seven layers of Bidirectional LSTM [17] and one fully connected layer. We optimize the Connectionist Temporal Classification (CTC) loss function [13], which was specifically designed for ASR systems. The model gets as input raw spectrograms (extracted using a window size of 25ms, frame-size of 10ms and Hamming window), and outputs a transcript.

A standard evaluating metric in speech recognition is the Word Error Rate (WER) or Character Error Rate (CER). These metrics were derived from the Levenshtein Distance [23], which is the number of substitutions, deletions, and insertions divided by the target length. The model achieves 12% Word Error Rate and 1.5% Character Error Rate on the Librispeech dataset [27], with no additional language modeling. In order to use Houdini for attacking an end-to-end ASR model, we need to get $g_\theta(x, y)$ and $g_\theta(x, \hat{y})$, which are the scores for predicting $y$ and $\hat{y}$ respectively. Recall, in speech recognition, the target $y$ is a sequence of characters. Hence, the score $g_\theta(x, \hat{y})$ is the sum of all possible paths to output $\hat{y}$. Fortunately, we can use the forward-backward algorithm [30], which is at the core of the CTC, to get the score of a given label $y$.

**ABX Experiment**   We generated 100 audio samples of adversarial examples and performed an ABX test with about 100 humans. An ABX test is a standard way to assess the detectable differences between two choices of sensory stimuli. We present each human with two audio samples A and B. Each of these two samples is either the legitimate or an adversarial version of the same sound. These two samples are followed by a third sound X randomly selected to be either A or B. Next, the human must decide whether X is the same as A or B. For every audio sample, we executed the ABX test with at least nine (9) different persons. Overall, Only 53.73% of the adversarial examples could be distinguished from the original ones by the humans (the optimal ratio is 50%). Therefore the generated examples are not statistically significantly distinguishable by a human ear. Subsequently, we use such indistinguishable adversarial examples to test the robustness of ASR systems.

**Houdini vs Probit Loss**   Houdini and the Probit loss [19] are tightly related. We also initially experimented with Probit but decided not to consider it further because: (1) Houdini is computationally more efficient. It requires only one forward-backward pass to generate adversarial examples while Probit needs several times more passes as it must average many networks to reduce the variance

|        | $\epsilon = 0.3$ |     | $\epsilon = 0.2$ |     | $\epsilon = 0.1$ |     | $\epsilon = 0.05$ |     |
|--------|------|-----|------|-----|------|-----|------|-----|
|        | WER  | CER | WER  | CER | WER  | CER | WER  | CER |
| CTC    | 68   | 9.3 | 51   | 6.9 | 29.8 | 4   | 20   | 2.5 |
| Houdini| 96.1 | 12  | 85.4 | 9.2 | 66.5 | 6.5 | 46.5 | 4.5 |

Figure 5: CER and WER in (%) for adversarial examples generated by both CTC and Houdini.

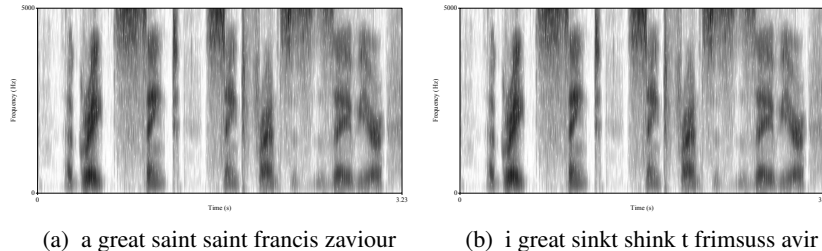

(a) a great saint saint francis zaviour          (b) i great sinkt shink t frimsuss avir

Figure 6: The model's output for each of the spectrograms is located at the bottom of each spectrogram. The target transcription is: `A Great Saint Saint Francis Xavier.`

of the gradients. (2) The "adversarial" examples generated with Probit are not adversarial in the sense that they are easily distinguishable from the original examples by a human. This is due to the noise (added to the parameters when computing the gradients with Probit) which seems to add white noise to the sound files. We calculated the character error rates (CER) and the percentage of examples that could be distinguished from original ones by a human (best is 50) for Houdini and Probit on the speech task. We used a perturbation of magnitude $\epsilon = 0.05$ and sampled 20 models for Probit (therefore 20x more computationally expensive than Houdini). In our results, while Probit and Houdini respectively achieve a CER of $5.97$ and $4.50$, adversarial examples generated with Probit are perfectly distinguishable by human ($100\%$) in comparison to those generated with Houdini ($53.73\%$).

**Untargeted Attacks**    In the first experiment, we compare network performance after attacking it with both Houdini and CTC. We generate two adversarial example, from each of the loss functions (CTC and Houdini), for every samples from the clean test set of Librispeech (2620 speech segments). We have experienced with a set of different distortion levels, using the $\ell_\infty$ norm and WER as $\ell$. For all adversarial examples, we use $\hat{y} = $ "Forty Two", which is the "Answer to the Ultimate Question of Life, the Universe, and Everything." Results are summarized in 5. Notice that Houdini causes a bigger decrease regarding both CER and WER than CTC for all the distortions values we have tested. In particular, for small adversarial perturbation ($\epsilon = 0.05$) the word error rate (WER) caused by an attack with Houdini is 2.3x larger than the WER obtained with CTC. Similarly, the character error rate (CER) caused by an Houdini-based attack is 1.8x larger than a CTC-based one. Fig. 6 shows the original and adversarial spectrograms for a single speech segment. (a) shows a spectrogram of the original sound file and (b) shows the spectrogram of the adversarial one. They are visually undistinguishable.

**Targetted Attacks**    We push the model towards predicting a different transcription iteratively. In this case, the input to the model at iteration $i$ is the adversarial example from iteration $i - 1$. Corresponding transcription samples are shown in Table 7. We notice that while setting $\hat{y}$ to be phonetically far from $y$, the model tends to predict wrong transcriptions but not necessarily similar to the selected target. However, when picking phonetically close ones, the model acts as expected and predict a transcription which is phonetically close to $\hat{y}$. Overall, *targetted attacks seem to be much more challenging when dealing with speech recognition systems than when we consider artificial visual systems such as pose estimators or semantic segmentation systems.*

**Black box Attacks**    Lastly, we experimented with a black box attack, that is attacking a system in which we do not have access to the models' gradients but only to its predictions. In Figure 8 we show few examples in which we use Google Voice application to predict the transcript for both original

| Manual Transcription | Adversarial Target | Adversarial Prediction |
|---|---|---|
| a great saint saint Francis Xavier | a green thank saint frenzier | a green thanked saint fredstus savia |
| no thanks I am glad to give you such easy happiness | notty am right to leave you soggy happiness | no to ex i am right like aluse o yve have misser |

Figure 7: Examples of iteratively generated adversarial examples for targetted attacks. In all case the model predicts the exact target transcription of the original example. Targetted attacks are more difficult when the speech segments are phonetically very different.

```
Groundtruth Transcription:
The fact that a man can recite a poem does not show he remembers
any previous occasion on which he has recited it or read it.

G-Voice transcription of the original example:
The fact that a man can decide a poem does not show he
remembers any previous occasion on which he has work cited or read it.

G-Voice transcription of the adversarial example:
The fact that I can rest I'm just not sure that you heard there is any
previous occasion I am at he has your side it or read it.
```

```
Groundtruth Transcription:
Her bearing was graceful and animated she led her son by the hand and
before her walked two maids with wax lights and silver candlesticks.

G-Voice transcription of the original example:
The bearing was graceful an animated she let her son by the hand and
before he walks two maids with wax lights and silver candlesticks.

G-Voice transcription of the adversarial example:
Mary was grateful then admitted she let her son before the walks
to Mays would like slice furnace filter count six.
```

Figure 8: Transcriptions from Google Voice application for original and adversarial speech segments.

and adversarial audio files. The original audio and their adversarial versions generated with our DeepSpeech-2 based model are not distinguishable by human according to our ABX test. We play each audio clip in front of an Android based mobile phone and report the transcription produced by the application. As can be seen, while Google Voice could get almost all the transcriptions correct for legitimate examples, it largely fails to produce good transcriptions for the adversarial examples. *As with images [32], adversarial examples for speech recognition also transfer between models.*

## 7 Conclusion

We have introduced a novel approach to generate adversarial examples tailored for the performance measure unique to the task of interest. We have applied Houdini to challenging structured prediction problems such as pose estimation, semantic segmentation and speech recognition. In each case, Houdini allows fooling state of the art learning systems with imperceptible perturbation, hence extending the use of adversarial examples beyond the task of image classification. *What the eyes see and the ears hear, the mind believes.* (Harry Houdini)

**Acknowledgments** The authors thank Alexandre Lebrun, Pauline Luc and Camille Couprie for valuable help with code and experiments. We also thank Antoine Bordes, Laurens van der Maaten, Nicolas Usunier, Christian Wolf, Herve Jegou, Yann Ollivier, Neil Zeghidour and Lior Wolf for their insightful comments on the early draft of this paper.

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
