[Reviews · NeurIPS 2017]

Reviewer 1



In the adversarial machine learning literature, most of research has been focusing on methods for fooling and defending image classifiers. This the paper instead proposes to generate adversarial examples for machine learning models for structured prediction problems. The white-box attacks to these image classifiers generate adversarial examples by backgpropagating the gradient from a differentiable classification loss function. The authors instead propose a loss function (called Houdini) that is surrogate to the actual task loss of the model that we are interested in breaking. The paper is interesting to the community in that it shows empirical lessons in fooling a set of real-world problems (ASR, human pose estimation, semantic segmentation) different than image classification that has been well studied. However, the overall finding that fooling these models is possible is not totally surprising. The demonstration of a common surrogate loss (Houdini) that could be applied to multiple applications is important. My questions are: - What are the key differences between Houdini surrogate loss and the probit surrogate loss in [18,19] (Keshet et al)? Is the main difference in the gradient approximation where the authors instead use Taylor approximation? (which basically means that the main advantage of Houdini over probit is computation time?) - What is the main conclusion between direct maximizing task loss vs maximizing Houdini loss (which is the task loss times the stochastic component) in the context of adversarial ML? My takeaways from Table 1 and Table 2 is the two losses perform similarly. Could you please clarify? At the moment, I vote for borderline "Marginally above acceptance threshold". But I'm willing to change my decision after hearing from the authors.

Reviewer 2



This paper presents a way to create adversarial examples based on a task loss (e.g. word error rate) rather than the loss being optimized for training. The approach is tested on a few different domains (pose estimation, semantic segmentation, speech recognition). Overall the approach is nice and the results are impressive. My main issues with the paper (prompting my "marginal accept" decision) are: - The math and notation is confusing and contradictory in places, e.g. involving many re-definitions of terms. It needs to be cleaned up. - The paper does discuss enough whether optimizing the task loss with Houdini is advantageous against optimizing the surrogate loss used for training. It seems in some cases it is clearly better (e.g. in speech recognition) but the scores given in the image tasks do not really show a huge difference. There also aren't any qualitative examples which demonstrate this. Overall I would be happy to see this paper accepted if it resolves these issues. Specific comments: - There are various typos/grammatical errors (incl. a broken reference at line 172); the paper needs a proofread before publication. - "we make the network hallucinate a minion" What is a minion? - "While this algorithm performs well in practice, it is extremely sensitive to its hyper-parameter" This is a strong statement to make - can you qualify this with a citation to work that shows this is true? - "\eps \in R^d is a d-dimensional..." I don't see you defining that \theta \in R^d, but I assume that's what you mean; you should say this somewhere. - Some of your terminology involves redefinitions and ambiguity which is confusing to me. For example, you define g_\theta(x) as a network (which I presume outputs a prediction of y given x), but then you later define g_\theta(x, y) as the "score output by the network for an example". What do you mean by the "score output by the network" here? Typically the network outputs a prediction which, when compared to the target, gives a score; it is not a function of both the input and the target output (you have defined y as the target output previous). You seem to be defining the network to include the target output and some "score". And what score do you mean exactly? The task loss? Surrogate loss? Some other score? It seems like maybe you mean that for each possible output y (not a target output) given the intput x, the network outputs a score, but this involves a redefinition of y from "target output" to "possible output". You also define y as the target, but y_\theta(x) as a function which produces the predicted target. - "To compute the RHS of the above quantity..." Don't you mean the LHS? The RHS does not involve the derivative of l_H. - The arguments of l_H have changed, from \theta, x, y to \hat{y}, y in equation (9). Why? - "Convergence dynamics illustrated in Fig. 4 " - Figure 4 does not show convergence dynamics, it shows "most challenging examples for the pose re-targeting class..." - Suggestion: In image examples you show zoomed-in regions of the image to emphasize the perturbations applied. Can you also show the same zoomed-in region from the original image? This will make it easier to compare. - "Notice that Houdini causes a bigger decrease in terms of both CER and WER than CTC for all tested distortions values." I think you mean bigger increase. - In Table 5 it's unclear if "manual transcription" is of the original speech or of the adversarially-modified speech. - In listening to your attached audio files, it sounds like you are not using a great spectrogram inversion routine (there are frame artefacts, sounds like a buzzing sound). Maybe this is a phase estimation issue? Maybe use griffin-lim? - Very high-level: I don't see how you are "democratizing adversarial examples". Democratize means "make (something) accessible to everyone". How is Houdini relevant to making adversarial examples accessible to everyone? Houdini seems more like a way to optimize task losses for adversarial example generation, not a way of making examples avaiable to people.